# Breeding Peaches for Brown Rot Resistance in Embrapa

**Maximiliano Dini** [1,2,*] [ID]**, Maria do Carmo Bassols Raseira** [3]**, Silvia Scariotto** [4] **and Bernardo Ueno** [3]

1 Programa de Pós-Graduação em Agronomia, Faculdade de Agronomia Eliseu Maciel, Universidade Federal de Pelotas, Capão do Leão 96010-900, Rio Grande do Sul, Brazil
2 Instituto Nacional de Investigación Agropecuaria (INIA), Estación Experimental INIA Las Brujas, Ruta 48 km 10, Rincón del Colorado, Canelones 90100, Uruguay
3 Embrapa Clima Temperado, Empresa Brasileira de Pesquisa Agropecuária, BR 392, km 78, Pelotas 96010-971, Rio Grande do Sul, Brazil
4 Campus Pato Branco, Universidade Tecnológica Federal do Paraná, Via do Conhecimento, km 1, Pato Branco 85503-390, Paraná, Brazil
* Correspondence: mdini@inia.org.uy; Tel.: +598-96256299

**Abstract:** Brown rot, caused by *Monilinia* spp., is the main stone fruit disease. Major efforts to detect sources of resistance are being applied by several breeding programs worldwide. The main objective of this study was to seek sources of brown rot resistance, as well as to study the segregation, estimate the heritability, verify the possible existence of the maternal effect, and estimate the genetic advances. For this purpose, 20 parents and 303 seedlings, representing 16 breeding families, and 'Bolinha' (control) have been phenotyped for fruit reaction to brown rot using wounded and non-wounded inoculation procedures in 2015–2016, 2016–2017, and 2017–2018 growing seasons. Wounded fruits were very susceptible to brown rot incidence, however, the incidence and severity of non-wounded fruits showed high variability among the evaluated genotypes. Conserva 947 and Conserva 1600 and their progeny, had lower disease incidence and severity than most of the evaluated genotypes. Genetic gain estimation was −5.2 to −30.2% (wounded fruits) and between −15.0 to −25.0% (non-wounded fruits) for brown rot resistance. Selected genotypes were equal to or better than 'Bolinha' in relation to brown rot resistance, with several of them far superior in fruit quality than 'Bolinha', demonstrating the progress of the Embrapa Peach Breeding Program.

**Keywords:** *Prunus persica* (L.) Batsch; *Monilinia fructicola* (Winter) Honey; genetic resistance; progeny segregation; genetic advance; lesion; sporulation

## 1. Introduction

Peach (*Prunus persica* L. Batsch) breeding programs started in Brazil, in the early 1950s, in the Instituto Agronomico de Campinas, São Paulo State. A few years later a similar program started in the Rio Grande do Sul State. The latter was first coordinated by the Agricultural Secretary of Rio Grande do Sul. However, in the early 1960s, the program was moved to Pelotas (Rio Grande do Sul), and then coordinated by a federal institution. With the advent of Empresa Brasileira de Pesquisa Agropecuária (Embrapa) in 1973, the program was not only preserved but also increased [1].

The Rio Grande do Sul State is the main peach producer, with about 64% of the Brazilian production, occupying more than 11,4 thousand hectares, in the year 2019. Pelotas, where the Embrapa Peach Breeding is located, stands out as the largest national producer of peaches, accounting for more than 11% of the country's total production [2]. The area is characterized by very humid weather and mild temperatures, so very favorable to fungus diseases. Among them, brown rot, caused by *Monilinia fructicola* (Winter) Honey, is by far the most important disease and for that reason, one of the top priorities of the Embrapa Peach Breeding Program is the search for *M. fructicola* resistance [1]. Under the above mentioned conditions, the disease symptoms may be visible in 48 h after infection. Brown

rot damages can occur from flowering to post-harvest. The main symptoms are blossom blight, cankers in branches, and brown rot lesions in fruits [3–5].

The use of genetic resistance has been limited in commercial orchards, since commercial peach cultivars immune to brown rot are still unavailable. However, there are significant differences in susceptibility among available genotypes [3]. Brown rot control is mainly done by fungicide sprays [6]. Thus, genetic resistance is a priority in many breeding programs worldwide, mainly due to the emergence of fungus isolates resistant to fungicides [7–11] and the increase in environmental concerns and workers' and consumers' health [1,12,13].

The Brazilian cultivar Bolinha shows higher levels of resistance in fruits than most cultivars, however, its poor fruit quality, together with problems of premature fruit fall, discourage its commercial cultivation [14–18].

Fruit resistance to brown rot is quantitative and polygenic trait [13,19,20]. This type of resistance slows the development of the epidemic in orchards, in spite of the genotype being susceptible to the disease [21,22].

Knowledge of the genetic, phenotypic, and environmental parameters that directly or indirectly influence the characters of economic importance in a crop is fundamental for breeding program guidance [23]. Given the limited information on peach brown rot resistance in Brazilian genotypes and with the intention of contributing to the genetic improvement of this crop, the objectives of this study were to evaluate the distribution of brown rot resistance in fruits of different populations; verify the possible existence of maternal effect for this character; estimate the heritability; identify genotypes with higher levels of resistance; estimate the genetic advance for this character in the Embrapa Peach Breeding Program.

## 2. Materials and Methods

### 2.1. Plant Material

The study was developed at Embrapa Clima Temperado, in Pelotas, Rio Grande do Sul, Brazil (Lat. 31°40′ S, Long. 52°26′ W, alt. 57 m asl.) in the 2015–2016, 2016–2017 and 2017–2018 seasons. The fruit reaction to brown rot was tested in 16 $F_1$ progenies of the Embrapa Peach Breeding Program, including ten progenies from five reciprocal crosses. Fruits of individual seedlings and their parents were evaluated (Table 1).

**Table 1.** Progeny identification, parents, and number of seedlings of each progeny, in the Peach Breeding Program at Embrapa Clima Temperado, Pelotas, Rio Grande do Sul, Brazil.

| Progeny | Parent ♀ | Parent ♂ | N° Seedlings |
|---|---|---|---|
| 2008.159 | Conserva 1526 | 'Cerrito' | 7 |
| 2009.38 | 'Cerrito' | Conserva 1526 | 23 |
| 2012.26 | Cascata 1055 | 'Chimarrita' | 18 |
| 2012.43 | 'Chimarrita' | Cascata 1055 | 25 |
| 2012.49 | Conserva 672 | Conserva 1526 | 18 |
| 2012.61 | Conserva 1526 | Conserva 672 | 7 |
| 2012.52 | Conserva 947 | Conserva 1600 | 17 |
| 2012.66 | Conserva 1600 | Conserva 947 | 12 |
| 2012.68 | Conserva 1662 | 'Maciel' | 24 |
| 2012.88 | 'Maciel' | Conserva 1662 | 17 |
| 2012.31 | Cascata 1359 | Cascata 1577 | 31 |
| 2012.46 | Chorão | 'Maciel' | 25 |
| 2012.99 | Necta 506 | 'Sunmist' | 20 |
| 2012.107 | Necta 532 | Necta 480 | 25 |
| 2012.111 | Necta 540 | 'Morena' | 25 |
| 2012.114 | 'Rubimel' | TX2D163 | 21 |

Seedlings were planted spaced 0.5 m apart on the row and 5 m between rows. The parent trees were planted in triplicate in a work collection of Embrapa, spaced 2 m between trees and 5 m between rows.

A total of 303 seedlings, 20 parents (cultivars and advanced selections), and the cultivar Bolinha were evaluated for their reaction to brown rot on W and NW fruits.

### 2.2. Experimental Design and Treatments

The experimental design was completely randomized, and each genotype was considered one treatment. Five wounded (W) (2015–2016, 2016–2017, and 2017–2018 seasons) and five non-wounded (NW) fruits (2016–2017 and 2017–2018 seasons) per seedling were inoculated. In the same seasons, 10 inoculated fruits (five W and five NW), per each one of the three parents clones were evaluated. The Bolinha cultivar was used as a standard for low brown rot susceptibility [14,15].

### 2.3. Pathogen Culture, Conidia Production, and Inoculation

The procedure adopted for inoculations was the same described by Dini et al. [24]. Briefly, the fungus isolate was obtained from four peach orchards in Embrapa Clima Temperado. For the wounded inoculations (penetration of 1 mm into the fruits) a 100 μL syringe coupled to a repeating dispenser $50\times$ (Hamilton®) was used. In both treatments (W and NW) 10 μL suspension of *M. fructicola* ($2.5 \times 10^4$ conidia mL$^{-1}$) and Tween-80® ($0.1$ g L$^{-1}$) was used, followed by incubation at $23 \pm 1$ °C and 12 h photoperiod for three days [16,18,19,24–27].

### 2.4. Brown Rot Evaluation

Incidence and severity 72 h after inoculation (HAI) were evaluated according to Dini et al. [24]. Brown rot incidence (BRI) and sporulation presence (SPP) were calculated as the percentage of fruits that presented symptoms and sporulation of *M. fructicola*, respectively; lesion diameter (LD) and sporulation diameter (SPD) were evaluated through two perpendicular measurements per fruit in the equatorial region; lesion area (LA) and lesion sporulation (SPA) were calculated and expressed as the percentage of fruit that was affected by the lesion and sporulation, respectively. Thus, fruit size was considered in each measurement making them comparable, which was important since fruit size was very variable among genotypes.

### 2.5. Statistical and Genetic Analysis

To evaluate the progeny segregation and to test for the possibility of maternal effect, relative frequency histograms were constructed with the severity data. The maternal effect was tested by comparing the $F_1$ progeny with the $F_1$ reciprocal progeny, by the t-test ($p < 0.05$) [28].

Broad-sense heritability ($H^2$) was estimated according to Dini et al. [18] for resistance to brown rot in fruits. The calculation was based on the data obtained in all seasons evaluated for W and NW fruits, and the estimated environmental variance ($\hat{\sigma}_e^2$) divided by three and two, respectively (environments number = seasons of evaluation). Narrow-sense heritability ($h^2$) was estimated from the regression of the average of parental phenotypic values vs. offspring phenotypic values [29–31].

Genetic advance (GA%), also called response to selection or genetic progress was estimated, and expressed in percentage of the population mean, using the formula:

$$GA\% = \frac{(\overline{X}_s - \overline{X}_o)\ h^2}{\overline{X}_o} * 100 \tag{1}$$

where: $\overline{X}_s$ mean of the selected genotypes; $\overline{X}_o$ original mean (base population); $\overline{X}_s - \overline{X}_o = SD$, selection differential; $h^2$, narrow-sense heritability [29]. 'Bolinha' was then used as a reference for the selection of the best genotypes.

## 3. Results

Evaluation of wounded fruits showed that all genotypes were susceptible to BRI, varying between 92–100% in the parents and between 83–99% in the progeny (Figure 1A). However, 22 genotypes showed a BRI of less than 80%, being most of them on progeny 2012.52 (4 seedlings), 2012.66 (5 seedlings), and 2012.68 (5 seedlings).

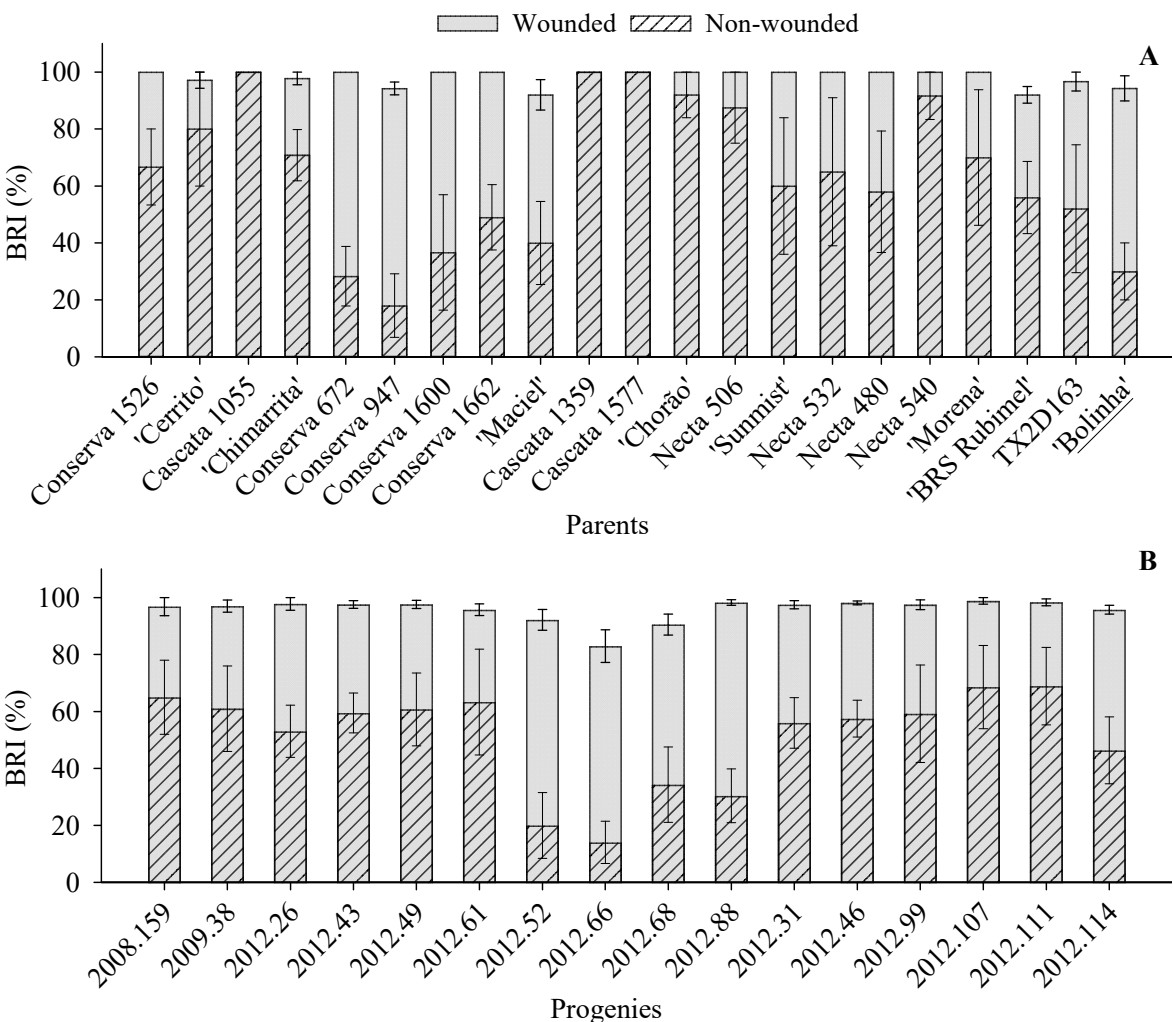

**Figure 1.** Brown rot incidence (BRI) on wounded and non-wounded (W and NW) fruits of parents (**A**) and progeny (**B**) average for three and two seasons (2015–2016; 2016–2017; 2017–2018 and 2016–2017; 2017–2018, respectively). 'Bolinha' was included as control. Embrapa Peach Breeding Program, Pelotas, Rio Grande do Sul, Brazil.

When inoculation was made on NW fruits, there was a wide variability, between 18–100% among parent plants and between 14–69% in the progeny (Figure 1B). The progeny with the lowest means were 2012.52 (20%), 2012.66 (14%), 2012.68 (34%) and 2012.88 (30%). A total of 43 seedlings presented less than 20% BRI, and most of them are part of the aforementioned progeny (5, 5, 6, and 4 seedlings, respectively). It should be noted that the progeny 2012.52 and 2012.66 are a product of a controlled cross between Conserva 947 and Conserva 1600, and in turn, the progeny 2012.68 and 2012.88 are a product of the cross between Conserva 1662 and 'Maciel', parents who presented, together with Conserva 672, the lowest means. In addition, the progeny 2012.114 stood out with six seedlings with less than 20% of BRI, and all seedlings produced fruits with a marked pilosity.

Under the W treatment the seedlings the LA followed approximately normal distribution. On the other hand, NW treatment exhibited seedling distribution like Lognormal

or Weibull, concentrated in classes of lower LA (0–10 and 10–20% of the area affected by lesion) (Figure 2).

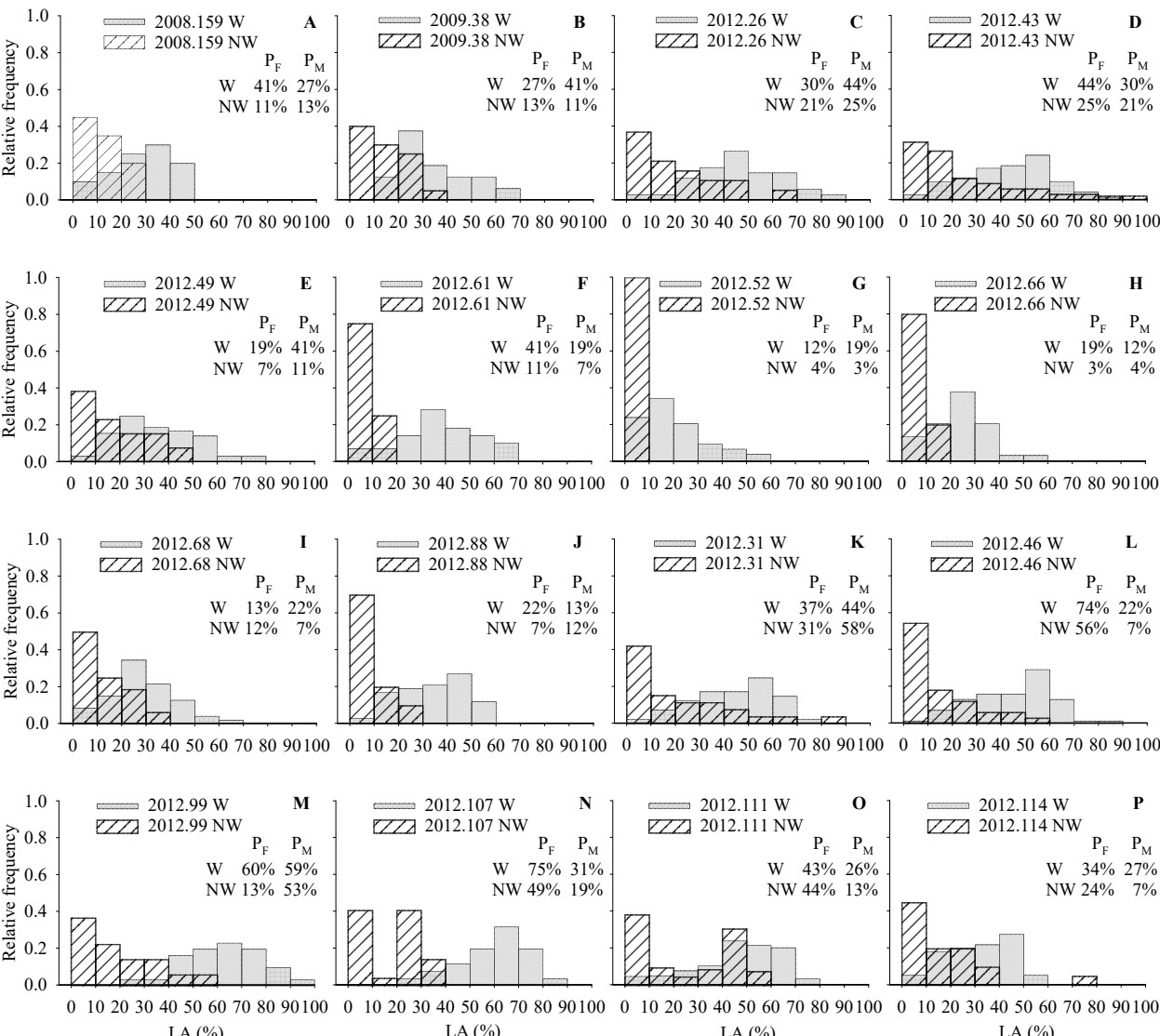

**Figure 2.** Relative frequency histograms of progeny of 16 families (**A**–**P**) classified by lesion area (LA) expressed as percentage of the fruit that was affected by the brown rot lesion, on wounded (W) and non-wounded (NW) fruits. The mean values of female ($P_f$) and male ($P_m$) parents for each population are indicated. Embrapa Peach Breeding Program, 2015–2016, 2016–2017 and 2017–2018 growing seasons, Pelotas, Rio Grande do Sul, Brazil. Progeny: 2008.159 (**A**), 2009.38 (**B**), 2012.26 (**C**), 2012.43 (**D**), 2012.49 (**E**), 2012.61 (**F**), 2012.52 (**G**), 2012.66 (**H**), 2012.68 (**I**), 2012.88 (**J**), 2012.31 (**K**), 2012.46 (**L**), 2012.99 (**M**), 2012.107 (**N**), 2012.111 (**O**) and 2012.114 (**P**).

The 2012.52 and 2012.66 progeny (Figure 2G,H), grouped the highest percentage of seedlings in the first two categories of LA (58 and 35%, respectively) when the fruits were W, and under NW treatment all genotypes of these progeny, together with the 2012.61 progeny (Figure 2F), were grouped within the first two categories (0–10 and 10–20% of the area affected by lesion). The parents of 2012.52 and 2012.66 progeny (Conserva 947 and Conserva 1600) had the lowest means of LA for both W fruits (12 and 19% respectively) and NW treatment (4 and 3%, respectively). Conserva 672 selection, one of the parents of 2012.61 progeny, also presented low LA value (19 and 7%, for W and NW fruits, respec-

tively), as well as the parents of 2012.68 and 2012.88 progeny (Conserva 1662 and 'Maciel') of LA 13 and 22% (W) and 12 and 7% (NW), respectively (Figure 2I,J).

The 2012.26 and 2012.43 progeny (Figure 2C,D) and the 2012.99, 2012.107 and 2012.111 nectarine progeny (Figure 2L,M,N, respectively) were the most susceptible to brown rot on W fruits, with a high concentration of seedlings in the categories of 40 to 70% of the area affected by the lesion.

In relation to SPP, high variability was detected for both W and NW fruits, with intervals for the parents between 16–96% and 0–94%, respectively (Figure 3A), and a range of 14–69% and 0–35% for W and NW fruits of the progeny, respectively (Figure 3B). The 2012.52 and 2012.66 progeny had only 20 and 14% of SPP, respectively, on W fruits, while no seedlings with SPP on NW fruits were observed in both progenies. Secondly, the 2012.68 and 2012.88 progeny had less than 35% of SPP on W fruits and, the 2012.61 progeny, stood out with less than 11% of SPP on NW fruits. Among the parents, those with lower SPP mean for W and NW fruits were Conserva 947 and Conserva 1600 (parents of 2012.52 and 2012.66), Conserva 672 (parent of 2012.49 and 2012.61), and Conserva 1662 (parent of 2012.68 and 2012.88) (Figure 3A).

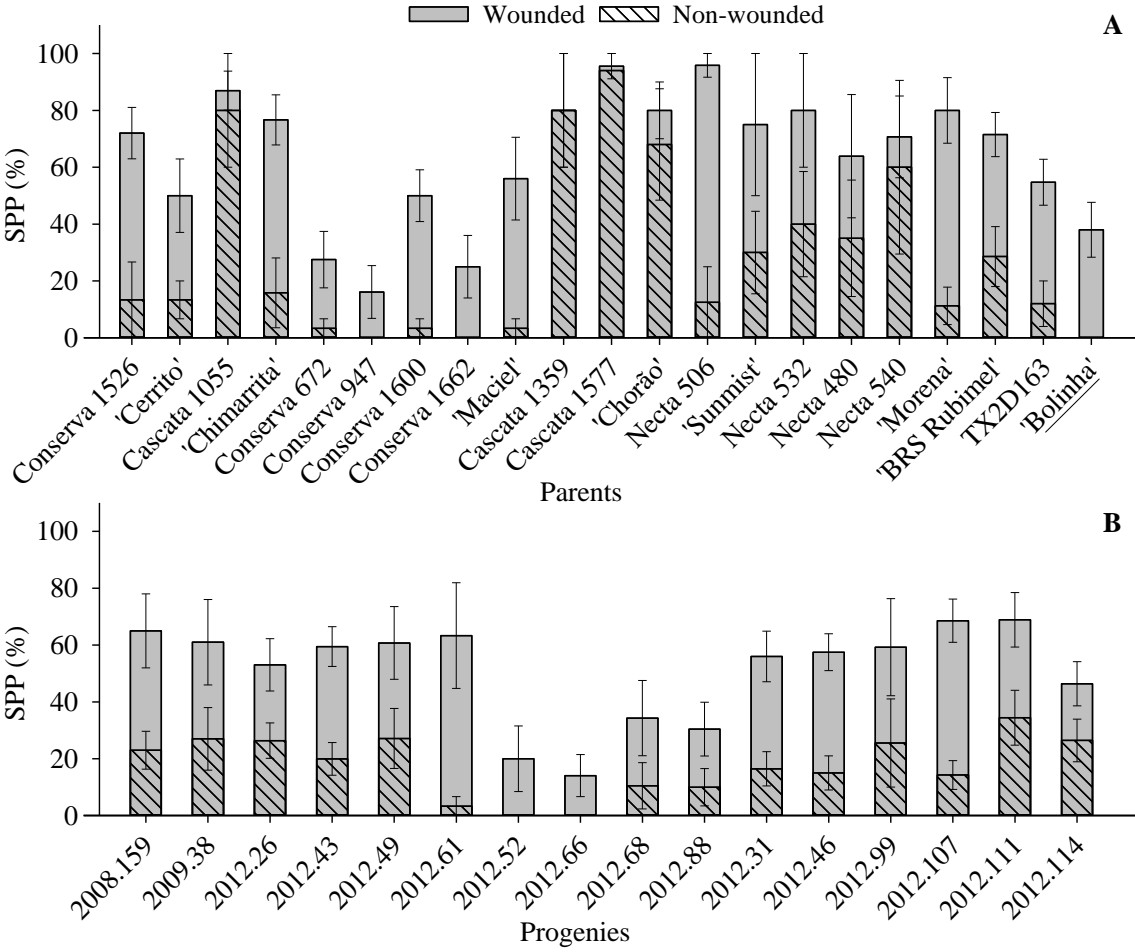

**Figure 3.** Sporulation presence (SPP) on wounded and non-wounded fruits of parents (**A**) and progeny (**B**) averages for three and two seasons (2015–2016; 2016–2017; 2017–2018 and 2016–2017; 2017–2018, respectively). 'Bolinha' was included as control. Embrapa Peach Breeding Program, Pelotas, Rio Grande do Sul, Brazil.

For SPA, on W and NW fruits, the seedlings followed approximately a Lognormal or Weibull distribution, with a high concentration in the lower categories, mainly the first one

(0–10% of the area affected by sporulation) (Figure 4). The only progeny that presented an approximately normal distribution were those of nectarines, on W fruits (Figure 4M–O).

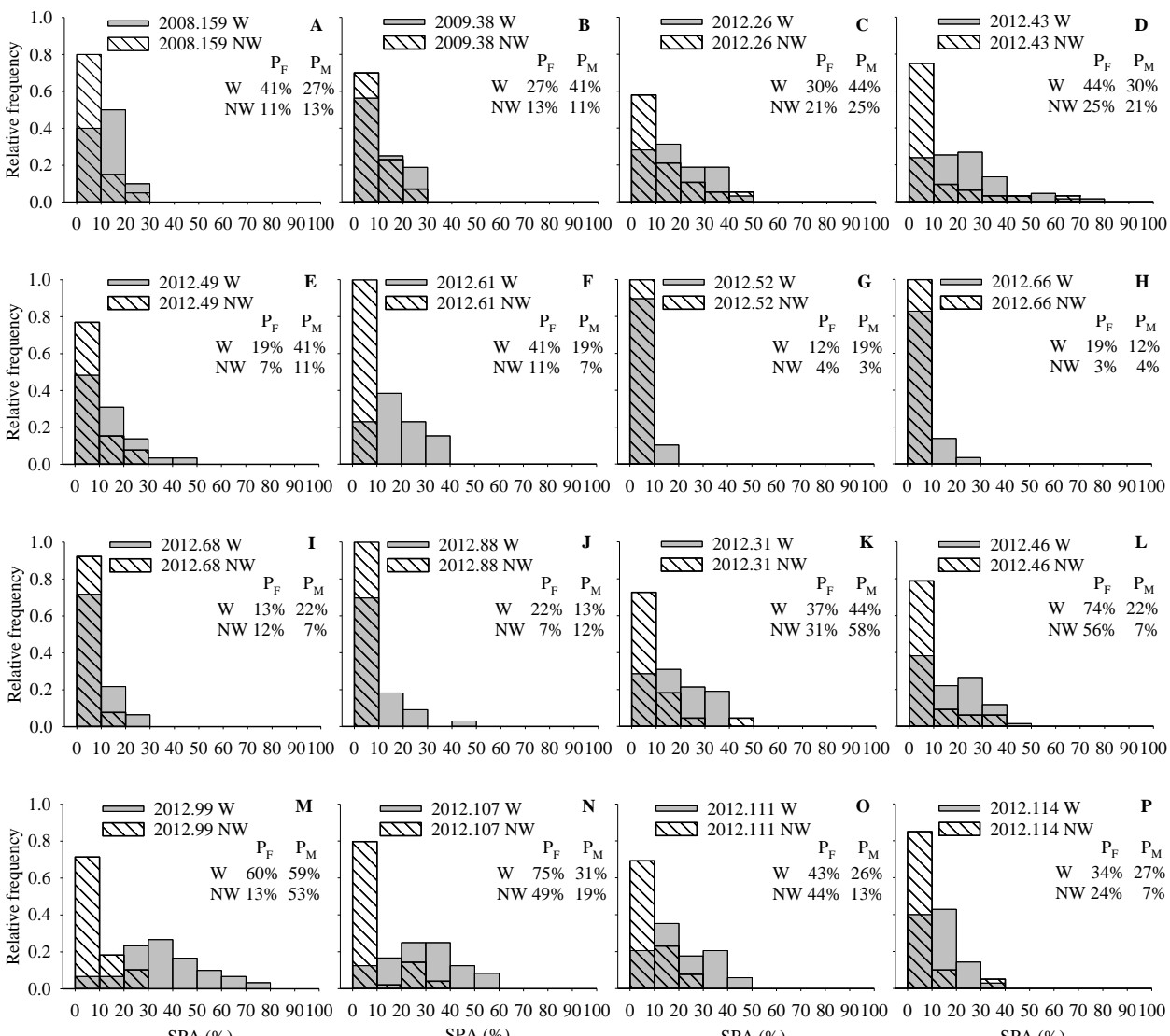

**Figure 4.** Relative frequency histograms of progeny of 16 families (**A–P**) classified by sporulation area (SPA) expressed as percentage of the fruit that was affected by the brown rot sporulation, on wounded (W) and non-wounded (NW) fruits. The mean values of female ($P_f$) and male ($P_m$) parents for each population are indicated. Embrapa Peach Breeding Program, 2015–2016, 2016–2017 and 2017–2018 growing seasons, Pelotas, Rio Grande do Sul, Brazil. Progeny: 2008.159 (**A**), 2009.38 (**B**), 2012.26 (**C**), 2012.43 (**D**), 2012.49 (**E**), 2012.61 (**F**), 2012.52 (**G**), 2012.66 (**H**), 2012.68 (**I**), 2012.88 (**J**), 2012.31 (**K**), 2012.46 (**L**), 2012.99 (**M**), 2012.107 (**N**), 2012.111 (**O**) and 2012.114 (**P**).

The maternal effect was analyzed (the contrast between the F1 progeny versus its reciprocal) for all the parameters (BRI, LD, LA, SPP, SPD, and SPA), and in all cases the differences were not significant ($p > 0.05$) indicating that there is no maternal effect involved in the transmission of this trait.

$H^2$ estimates were from medium to low, for the different parameters, ranging between 30.1–51.7% on W fruits and between 25.0–40.5% on NW fruits. The $h^2$ estimates varied depending on the year of evaluation, but the obtained means were between 18.3–41.6% for W fruits and between 18.7–33.9% for NW fruits (Table 2).

**Table 2.** Broad-sense ($H^2$) and narrow-sense ($h^2$) estimated heritability for incidence and severity of *M. fructicola* on wounded and non-wounded peach fruits, Embrapa Peach Breeding Program, Pelotas, Rio Grande do Sul, Brazil.

| Fruits | Traits * | H² (%) | h² (%) | | | |
|---|---|---|---|---|---|---|
| | | | 2015–2016 | 2016–2017 | 2017–2018 | Mean |
| Wounded | BRI | 42.5 | 35.8 | 37.6 | 31.1 | 34.8 |
| | LD | 51.7 | 47.1 | 42.1 | 35.5 | 41.6 |
| | LA | 44.7 | 33.5 | 41.1 | 25.5 | 33.4 |
| | SPP | 27.5 | 23.2 | 18.0 | 13.8 | 18.3 |
| | SPD | 41.2 | 33.0 | 39.3 | 35.9 | 36.1 |
| | SPA | 30.1 | 21.0 | 28.9 | 18.5 | 22.8 |
| Non-wounded | BRI | 26.7 | - | 20.2 | 17.1 | 18.7 |
| | LD | 40.5 | - | 34.6 | 30.1 | 32.4 |
| | LA | 29.8 | - | 28.6 | 20.2 | 24.4 |
| | SPP | 25.0 | - | 20.1 | 11.8 | 16.0 |
| | SPD | 31.1 | - | 29.3 | 19.9 | 24.6 |
| | SPA | 23.6 | - | 21.8 | 16.9 | 19.4 |

* BRI, brown rot incidence; LD, lesion diameter; LA, lesion area; SPP, sporulation presence; SPD, sporulation diameter; SPA, sporulation area.

Using the 'Bolinha' as the cut-off point for resistance selection (Figure S1), the genetic advances obtained by the Embrapa Peach Breeding Program were estimated. The selection differential (SD) is the subtraction of the selected genotypes mean ($\overline{X}_s$) minus the original mean ($\overline{X}_o$).

The GA% associated with each variable was estimated for W and NW fruits, separately (Table 3). The selected genotype number (SGN) was between 11 and 76, and GA% estimates were between −5.2 to −30.2% on W fruits. On the other hand, the SGN on NW fruits was between 53 and 93, and GA% estimates were between −15.0 and −27.3%.

**Table 3.** Genetic advancement by selection of genotypes better than or equal to 'Bolinha'.

| | Wounded | | | | | | Non-Wounded | | | | | |
|---|---|---|---|---|---|---|---|---|---|---|---|---|
| | BRI | LD | LA | SPP | SPD | SPA | BRI | LD | LA | SPP * | SPD * | SPA * |
| $\overline{X}_o$ | 96.3 | 29.5 | 39.7 | 60.6 | 14.6 | 15.9 | 54.0 | 12.8 | 16.6 | 19.4 | 4.5 | 5.4 |
| 'Bolinha' | 94.3 | 20.2 | 13.3 | 37.1 | 5.5 | 2.1 | 30.0 | 6.5 | 4.0 | 0 | 0 | 0 |
| $\overline{X}_s$ | 81.8 | 14.5 | 8.1 | 15.8 | 2.4 | 0.5 | 11.0 | 1.9 | 0.9 | 0 | 0 | 0 |
| SD | −14.5 | −15.0 | −31.6 | −44.8 | −12.2 | −15.5 | −43.0 | −10.9 | −15.7 | −19.4 | −4.5 | −5.4 |
| SGN | 58 | 21 | 11 | 76 | 63 | 41 | 53 | 59 | 56 | 93 | 93 | 93 |
| h² | 0.35 | 0.42 | 0.33 | 0.18 | 0.36 | 0.23 | 0.19 | 0.32 | 0.24 | 0.16 | 0.25 | 0.19 |
| GA (direct) | −5.0 | −6.2 | −10.5 | −8.2 | −4.4 | −3.5 | −8.1 | −3.5 | −3.8 | −3.2 | −1.1 | −1.0 |
| GA% | −5.2 | −21.1 | −26.5 | −13.5 | −30.2 | −22.1 | −15.0 | −27.3 | −22.8 | −16.4 | −25.0 | −18.6 |

BRI, brown rot incidence; SPP, sporulation presence; LD, lesion diameter; SPD, sporulation diameter; LA, lesion area; SPA, sporulation area; h², mean of narrow-sense heritability; SD, selection differential; SGN, selected genotypes number; $\overline{X}_s$, mean of the selected genotypes; $\overline{X}_o$, original mean (base population); GA (direct), direct response; GA%, genetic advance (selection response in percentage of the population mean); * 'Bolinha' value = 0.

Using the cultivar Bolinha as control, four seedlings (2012.52.17, 2012.52.2, 2012.66.11, and 2012.68.24) and a selection (Conserva 947) were identified, with better results in all variables of incidence and severity to brown rot (BRI, LD, LA, SPP, SPD, and SPA) on W fruits. In the case of NW fruits, 40 seedlings and three selections (Conserva 947, Conserva 1600, and Conserva 672) had equal or better results than 'Bolinha'.

## 4. Discussion

One of the top priorities of the Embrapa Peach Breeding Program is disease resistance, which is justified by the weather conditions of the area that is one the most important in Brazil, for stone fruit production. The climate favors also the incidence of insect pests

causing damage to the fruit epidermis. Thus, the tests using W fruits were especially important and proved to be suitable to identify resistance sources.

Exploring sources of brown rot resistance in an interspecific population of almond × peach, Baró-Montel et al. [13] reported similar results to those of the present study regarding BRI, where close to 100% of W fruits developed the disease, and NW fruits varied between 0–80% of BRI. On the other hand, working with NW fruits, BRI averages were reported between 60–100% [26] and between 50–100% [27]. These three studies performed the evaluation at 120 HAI and in the present study evaluations were made at 72 HAI.

In inoculation without wounding, the lower development of the disease may be due to the delay in fungus infection and/or in the lower probability of conidia having a successful penetration of fruit skin. Although visually the fruits appeared to be undamaged, there may be microcracks where the fungus could infect. In the case of *M. fructicola*, it has been reported that the wound is necessary and is the main gateway for infection [25,32–34]. Thus, the conidia number that would have success in the infection, in the case of inoculation with a wound would be much greater in relation to inoculation without a wound. For this reason, the standard error bars are much smaller in the evaluations with injury when compared to the uninjured fruit evaluations, both for BRI in the parents and in the progeny (Figure 1). This indicates that the sample means were more reliable when the wound was used, and this methodology is recommended, since microcracks of diverse origin can exist and are highly associated with the environmental conditions and not with the genotype, presenting a disparity between the evaluated seasons [13,27,35,36].

The 2012.114 progeny, with 46% BRI on NW fruits, had six seedlings with less than 20% of BRI (Figure 1B). This progeny was characterized by producing fruits with a high pilosity, and this characteristic associated with unwounded inoculation can be considered a structural barrier, hindering the development and infection of *M. fructicola* conidia in the fruit [22,37]. However, in stone fruits, the subject is controversial, since there are reports where the trichomes favor the infection of *Monilinia* spp. [8,38,39]. These studies mention that microcracks at the base of the trichomes may result in entry points of the fungus. In addition, the high density and length of the trichomes in some genotypes of the progeny 2012.114 can have caused a failure in the technique of inoculation, since the drop with a conidial suspension of *M. fructicola* may have dried before coming into contact with the fruit skin.

In the 2012.107 and 2012.111 nectarine progeny, erratic behavior was observed when the fruits were evaluated without wounds (Figure 2N,L). Even though a large number of fruits had no BRI, it was not associated with a specific genotype. This can be explained by the presence of waxes associated with the nectarine's cuticle, which may be considered a structural barrier to fungal infection [33,35,37]. However, it can also be an error in the inoculation technique, because a more waxy surface can cause the drop containing the conidial suspension of *M. fructicola* to simply drain from the fruit or dry without having sufficient time of contact with the fruit skin, so fungus did not infect.

High variability related to sporulation variables such as SPP (Figure 3) and SPA (Figure 4), was detected both for W and NW fruits. In a study in which NW fruits were inoculated with *M. laxa*, and evaluated 120 HAI, Obi et al. [27] obtained averages were higher than in the present study (above 80%) of fruits with fungus sporulation (colonized), probably due to the longer incubation time and/or genetic differences. Genotypes that had rotted, but did not result in fungus sporulation, are of great epidemiological for the disease, reducing the secondary inoculum in orchard [5,22]. Among the more than 300 genotypes evaluated in this study, 23 (W) and 93 (NW) did not exhibit sporulation in any of their fruits, evaluated for three and two years, on W and NW fruits, respectively.

No maternal significant effect was detected in the inheritance of the characters associated with brown rot resistance. The use of reciprocal crosses is the simplest evidence of testing the maternal effect, since they would produce genetically similar but phenotypically different individuals [23,31], if there was indeed a significant maternal effect. In stone fruits,

there are rare cases where a possible maternal effect has been mentioned [40,41] or maternal inheritance [42–44] and has never been associated with resistance to pests or diseases.

All $H^2$ and $h^2$ estimates were from medium to low and, considering that the peach resistance to brown rot is quantitative [13,19,20], the parents selection based on their phenotype may be considered reasonable to not very effective. Studying the reaction to *M. fructicola* on wounded peach fruits from the Embrapa Peach Breeding Program's work collection, Scariotto [17] estimated the $H^2$ of LD and SPD at 50 and 13%, respectively. The first estimate was similar to the present study (51.7%), but the second estimate was considerably lower (41.2%). On the other hand, testing the response on unwounded peach fruits of Embrapa Peach Breeding Program populations [45] estimated the BRI $H^2$ at 64%, which is different from this study where for the same variable and under the same conditions it was 26.7%. Although the site where these experiments were done was the same, there were differences in methods, climatic conditions, and genotypes. Considering these heritability estimates (low to medium values), moderate genetic advance with great environmental influence can be expected due to selection for the characters associated with brown rot resistance [29].

The lowest GA% was associated with BRI on W fruits (−5.2%), due to the low variability associated with this character in the studied populations. GA% of −13.5 to −30.2%, were estimated for the other variables on W fruits. On the other hand, for NW fruits, GA% estimates were between −15.0 to −27.3%, and in the case of the selected genotypes with respect to sporulation, there were those that presented 0 (SPP, SPD, and SPA) as 'Bolinha'.

There were not find references, in the literature, regarding genetic advances (genetic gains or response to selection) in relation to disease resistance and/or pests in *Prunus* spp. Estimating the heritability and predicted selection response of quantitative traits in peach, Souza et al. [46] obtained predictions from 4.61–61.25% on fruit quality characteristics (fruit mass, soluble solids, acidity, fruit length, and others) and between 20.44–95.00% for phenological characters (full bloom, ripening and fruit development period). In the same way, Chandrababu and Sharma [47] studying the heritability and genetic gain in several characteristics of almond (*Prunus dulcis*), obtained values of genetic gain between 37.98–187.27% for growth and yield characters, 33.77–132.79% for flowering and fruiting characters, and 23.08–74.36% for fruit and kernel characters. Polygenic traits have low heritability and high environmental influence, and the genetic gain expected by selection based on the phenotype is generally moderate to low [29,48].

Among the five genotypes with the best results on W fruits, there were three seedlings and one selection related directly to 'Bolinha'. The advanced selection Conserva 947 was selected from a cross between 'Bolinha' × P60-22 (Mexican polen) and, the seedlings (2012.52.17, 2012.52.2, and 2012.66.11) originated from the cross between Conserva 947 × Conserva 1600 (2012.52), and its reciprocal (2012.66). Among the 43 selected genotypes with the best results on NW fruits, there were 11 genotypes related to 'Bolinha', five seedlings of 2012.52 progeny and five of 2012.66 progeny, in addition to selection Conserva 947. Besides these genotypes, the other progeny that participated with a higher number of selected genotypes were 2012.68 (one and six, on W and NW fruits, respectively) and 2012.88 (four, on NW fruits), progeny originated from the crossing between Conserva 1662 × 'Maciel' and their reciprocal cross, respectively. Another of the selected genotypes, Conserva 672 advanced selection, had previously been reported as being one of the genotypes with the best results on non-wounded fruits [45].

All selected genotypes with and without a wound, had equal or better results than 'Bolinha' for brown rot resistance in fruits and, several of them, mainly the selections (Conserva 947, Conserva 1600, and Conserva 672) have superior fruit quality to 'Bolinha', demonstrating the progress of the Embrapa Peach Breeding Program, not only regarding resistance to *M. fructicola*.

## 5. Conclusions

The studied populations present variability regarding brown rot resistance, without any evidence of maternal effect. The heritability of brown rot resistance in fruits is medium to low. Selection of genotypes based on phenotypes better or equal to 'Bolinha', allowed estimate of moderate to low genetic advances for brown rot resistance. The Embrapa Peach Breeding Program is achieving genetic advances in fruit resistance to *M. fructicola*, currently having several genotypes comparable to 'Bolinha' but with better fruit quality.

**Supplementary Materials:** The following supporting information can be downloaded at: https://www.mdpi.com/article/10.3390/agronomy12102306/s1. Figure S1: Hypothetical distribution of all evaluated genotypes from the 16 populations; selection was based on the incidence and severity means of 'Bolinha'. The selection differential (SD) is the subtraction of the selected genotype's mean ($\overline{X}_s$) minus the original mean ($\overline{X}_o$).

**Author Contributions:** Conceptualization, M.d.C.B.R. and B.U.; methodology, B.U., S.S. and M.D.; software, M.D.; validation, M.d.C.B.R. and B.U.; formal analysis, M.D.; investigation, M.D. and S.S.; resources, M.d.C.B.R.; data curation, M.D.; writing—original draft preparation, M.D.; writing—review and editing, M.D.; visualization, M.D. and M.d.C.B.R.; supervision, M.d.C.B.R. and B.U.; project administration, M.d.C.B.R.; funding acquisition, M.D. All authors have read and agreed to the published version of the manuscript.

**Funding:** This research was funded by National Agency for Research and Innovation (ANII-Uruguay) awarding the master's degree scholarship and the Coordination for the Improvement of Higher Education Personnel (CAPES-Brazil) awarding the first author's doctoral scholarship.

**Institutional Review Board Statement:** Not applicable.

**Informed Consent Statement:** Not applicable.

**Data Availability Statement:** Not applicable.

**Acknowledgments:** Authors thank Embrapa staff, especially Everton Pederzolli, Maicon Bönemann, and Gilberto Kuhn, and to the colleagues and friends who at some point helped with the practical work: Priscila Monalisa Marchi, Silvia Carpenedo, Carol Silveira, Julia Ritter, Diego Borges, and Leonardo Milech.

**Conflicts of Interest:** The authors declare no conflict of interest.

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
