# Peer review of "Breeding Peaches for Brown Rot Resistance in Embrapa"

_agronomy, doi:10.3390/agronomy12102306_

Round 1

Reviewer 1 Report

Very interesting study showing progression in breeding program towards achieving a goal of improved disease tolerance. Brown rot is a significant problem world wide and study such as this provides additional insight into the complexity of breeding for tolerance to this fungus. My major comments are regarding the English language and structure of the sentences that I tried to simplify in my review. General suggestion is to change wounded and non-wounded fruits to treatment and also could be simplified if W and NW acronyms were defined in the Material and methods to replace full spelling throughout the text.

Plural of progeny is progeny when it is used in non-counting and progenies is used if you mention actual number of the offspring e.g. 15 progenies of the 2012.52 but 2012.52 progeny. So Line 201 in Figure 4 caption is correct use of progenies although I would replace it with "... progeny of 16 families..." as in this context it might suggest 16 individual plants and not 16 crosses.

Particular corrections can be found in the attached reviewed text and I will also try to list them here in case my handwriting is hard to read.

Line 14 replace is with are in "...sources of resistance are being applied..."

Line 19 write seasons as 2015-16, 2016-17 and 2017-18 throughout the text

Lines 20-21 rewrite the sentence to read "..., however the incidence and severity of non-wounded fruits showed high..."

Line 22 replace progenies with either progeny or offspring and presented with had e.g. "...progeny, had lower disease..."

Line 24 replace presented with were and delete results

Line 25 replace being with with

Line 26 delete which and replace demonstrates with demonstrating, delete the before progress

Line 33 change coordinate to coordinated

Line 34 add in before early 1960's

Line 35 change them to then and coordinate to coordinated

Line 36 put Embrapa in brackets and remove brackets around the full name

Line 44 add by before far to read "by far"

Line 59 delete "The type of" and start the sentence with "Fruit resistance..."

Line 60 delete "the" in front of "orchard"

Line 63 what do you mean by culture, production?

Line 64 delete the before breeding

Line 66 add to between "were" and  "evaluate" to read "...were to evaluate..."

Line 77 delete the at the beginning of sentence before "individual"

Line 78 Table 1 - General comment remove quotations around cultivar names in tables

Line 80 Question, were families represented by a single tree for each progeny or there were multiple trees

Lines 81-82 insert "triplicate in" after "...planted in" to say "planted in triplicate in a work..." and delete the last sentence in the paragraph (line 82).

Lines 86-87 delete 'were used. Three clones of each parent" and add "parent" between "per" and "clone"  to read "... and ten fruits per parent clone, ..."

Line 91 and throughout the text when you reference the paper for some methods please add the authors followed by et. all before the reference number, e.g. Dini et al. [24]

Line 93 re arrange the sentence to read "For the wounded inoculations (penetration of 1 mm into the fruits) a 100 ul syringe coupled to a repeating dispenser 50x (Hamilton) was used."

Line 94 replace "cases" with treatments

LIne 95 correct spelling "suspention" should be "suspension"

Line 96 replace "in each fruit" with "followed by" to read: "...followed by incubation at 23+1 ºC..."

Line 101 delete "at"

Line 110 progeny not progenies

Line 121 replace "with" with "using"

Line 122 use "=" between the symbol and its meaning

Lines 149-152 rewrite the sentence to read " Under the wounded treatment the seedlings LA followed approximately normal distribution. On the other hands ,non-wounded treatment exhibited seedling distribution like Lognormal or Weibull, concentrated in classes of lower LA (..."

Line 163 delete "evaluated with" and add "ed" to "wound" to read wounded; replace "without" with "under" and change the following text to read  non-wounded treatment all genotypes of these progeny, ..."

Line 164 insert "grouped" between "were" and "within"

Line 166 change "progenies" to "progeny" and replace "presented" with "had"

Line 167 delete "fruits" and replace second "fruits" with "treatment"

Line 172 and 173 replace progenies with progeny

Line 179 delete" Highlight" and start the sentence with "The 2012.52...", change "progenies" to "progeny" and replace "with" with "had only"

Line 180 delete "bot progenies had"

Line 181 at the end of the sentence after "fruits" add "were observed in both progeny." Delete highlight and replace "with" with "had"

Line 182 replace "with" with "had"; delete "they" and replace "stand" with "stood"

Line 183 replace "who presented" with "with"

Line 195 delte "in" and "The"

Line 196 start sentence with "Only" and change "progenies" to "progeny"

Line 201 Figure 4 caption replace "16 progenies" with "progeny of 16 families"

Line 206 first sentence is written as material and method please re-write and combine with the following sentence that reports results.

Throughout the text please change "to" to "-" between the numbers e.g. line 212 change "30.1 to 51.7%" to 3"0.1-51.7%" except when the numbers are negative

Line 248 add “evaluations” between “study” and “were”

Line 250 change “wound” to “wounding”

Line 252 change “appear” to “appeared”

Line 253 change “can” to “could”

Line 263 change “presented” to”had”

Line 265 change “to” to “with” and rewrite sentence to say “”…., and this characteristics associated with unwounded inoculation can be considered a structural barrier,…

Line 275 replace “Besides” with “Even though”

Line 276 replace “that did not present” with “had no”; delete “but”

Line 277 delete “and”

Line 278 delete “this”

Line 281 delete “that”

Line 282 delete “the”

Line 286 insert “were” between “averages” and “higher”

Line 288 replace “presented” with “had” and “do” with “did”

Line 289 delete “available” and “the”

Line 291 delete “genotypes” and replace “present” with “exhibit” or “show” and replace “in” with “on”

Line 302 replace “from” with “considered”

Line 305 replace “like that of” with “similar to”

Lines 306-307 rewrite the sentence to read “On the other hand, testing the response on non-wounded peach fruits of Embrapa Peach Breeding Program populations [46] estimated the BRI H2 at 64%, which is different that this study where for the same variable and under the same conditions it was 23.7%.”

Line 309 replace “works” with experiments”

Lines 311-312 rewrite to read “…, moderate genetic advance with great environmental influence can be expected due to selection for the characters associated with brown rot resistance [29].”

Line318 replace “equal to” with “as”

Line 319 replace “It was not found “with “There are no”; replace “to” with “regarding”

Line 331 delete “selected”

Line334 replace “crossing” with “cross”

Line 343 replace “that presented” with “with”

Line 345 replace “presented” with “had”

Line 347 replace “are of” with “have”

Line 348 delete “which” and replace “demonstrates” with “demonstrating”

Line 351 delete “the” before “brown rot”

Line 354 delete “to” and add “of” between “estimate” and “moderate”

Lines 355-356 italicize brown rot scientific name

Line 356 replace “counting with” with “having

Line 358 change order so it reads “… of all evaluated genotypes”

Line 359 replace “,” with “;”, delete “the” and “made”

Line 360 replace “genotypes” with “genotype’s”

Line 371 check Data availability statement

Line 372 add at the beginning “Authors thank”

Line 373 replace “.” With “,” and add” and”

Line491 delete "made"

Author Response

Dear Reviewer 1,

Many thanks! for the complete correction and suggestions.

Best regards

Reviewer 2 Report

The experimental design and the results are well developed by the authors but their presentation in the manuscript must be improved.

English language and style must be improved.

Considering the results obtained in wounded and no wounded fruit, in my opinion, it could be better to focus the manuscript on the method set up and then, in order to save work done, describe the application of the set-up method on an experimental case. 

Some wording suggestion follows (they are only some examples)

Line 35: “them” should be “then”?

Line 43-45: “Among them, brown rot, caused by Monilinia fructicola (Winter) Honey, is far the most important and for that reason, the Embrapa Peach Breeding Program has the search for M. fructicola resistance as one of the priorities”. Text can be improved.

Line 45: Under these conditions…. Which conditions? Please, rephrase

Line 66: add “to” before “evaluate”, To verify…. To estimate…..

Line 100: wounds?

Line 133: “them” should be “there”

Line 179: “Highlight the 2012.52 and 2012.66 progenies”. Rephrase, it is unclear. Do you mean “It is to be highlighted”?

Line 227: “The selected genotypes number (SGN) was between 11 to 76” should be “The selected genotypes number (SGN) was between 11 and 76”

Line 235: “….had results equal or better than 'Bolinha” should be “had equal or better results than 'Bolinha”

Discussion: add some lines of introduction to the Discussion paragraph

Line 275-276: Basides a large number of fruits that did not present BRI, but it was not associated with a specific genotype. Please, rephrase

Author Response

Dear Reviewer 2,

Many thanks for the correction and suggestions.

Best regards.

Round 2

Reviewer 2 Report

The manuscript appears now improved.